# Effect of Residents-as-Teachers in Rural Community-Based Medical Education on the Learning of Medical Students and Residents: A Thematic Analysis

**DOI:** 10.3390/ijerph182312410

**Published:** 2021-11-25

**Authors:** Nozomi Nishikura, Ryuichi Ohta, Chiaki Sano

**Affiliations:** 1Community Care, Unnan City Hospital, 96-1 Iida, Daito-cho, Unnan City 699-1221, Shimane, Japan; ryuichiohta0120@gmail.com; 2Department of Community Medicine Management, Faculty of Medicine, Shimane University, 89-1 Enya cho, Izumo 693-8501, Shimane, Japan; sanochi@med.shimane-u.ac.jp

**Keywords:** residents-as-teachers, community-based medical education, rural community hospital, medical residents, medical students

## Abstract

Residents-as-teachers (RaT) is a theoretical framework emphasizing the significance of the similar learning background of teachers and learners. In Japan, community-based medical education (CBME) is a practical approach to teaching family medicine. This study aimed to investigate the impact and challenges of RaT on the learning of medical students and residents in CBME at a rural community hospital in Japan. Over the course of a year, the researchers conducted one-on-one interviews with three residents and ten medical students participating in family medicine training at the hospital. The interviews were recorded and transcribed verbatim. Grounded theory was used in the data analysis to clarify the findings. Three key themes emerged from the research: lack of educational experience with RaT, effectiveness of RaT, and challenges of RaT. Although participants were prejudiced against RaT, they felt its implementation could facilitate the establishment of beneficial relationships between learners and teachers. They were also able to participate in medical teams effectively. The findings suggest that the increased participation of senior doctors in RaT could strengthen its learning effects. RaT in rural CBME should be applied in various contexts, and its effectiveness should be further investigated both qualitatively and quantitatively.

## 1. Introduction

Medical residents are encouraged to develop their roles as teachers for mutual learning benefits for themselves and learners [1]. Residents-as-teachers (RaT) is a theoretical framework that emphasizes the significance of a similar knowledge base between teachers and learners, and mutual learning among non-professional teachers and learners in groups [2]. It is estimated that residents spend 20–25% of their time teaching students and peers [3,4], and medical students learn 30–85% of the curriculum in undergraduate medical education from residents [5]. Consequently, many medical students perceive residents as their most important and valuable clinical mentors or teachers [1,6]. Additionally, considering the diligence, time constraints, and workload of medical staff, there is a growing need for resident teachers. RaT may alleviate the pressure on overburdened medical teachers and preserve the quality of medical education in situations of limited medical and educational resources [7,8].

RaT has been reported to enhance residents’ self-efficacy in their professions and increase job satisfaction through mutual education in their training [9]. Additionally, it increases residents’ clinical competence [10] and professionalism through monitoring by their mentors and learners [11]. Furthermore, it provides learners with comfortable educational environments, enabling them to be more receptive to making mistakes and receiving constructive feedback from residents. This is because of the cognitive congruence between residents and learners, which allows the residents to engage in dialogue and discussion with them and to better explain the concepts. Moreover, they also share social congruence because of their similar social roles [12]. However, medical residents may perceive educational activities as burdensome, as the additional responsibilities of being educators may lead to delays in clinical work [13,14].

In Japan, the number of patients with multiple comorbidities is increasing due to the aging population [15]. To cope with complicated medical care and multimorbidity in an aging society, it is necessary to educate general practitioners and family physicians to deal with complex problems in collaboration with multiple professionals [16]. However, while medical students are required to study general practice, community-based comprehensive care, and primary care in medical universities, there may be inadequate education on primary care and family medicine [17]. 

In North America and Europe, first- and second-year medical students learn primarily in the classroom, whereas senior medical students and residents learn primarily through participation in clinical practice [18,19]. For senior medical students and residents, classroom learning, such as academic half days or noon conferences, remains an important but complementary aspect of their education and identity formation [20,21]. In Japan, first- and second-year medical students study various fields of the liberal arts as well as medicine. Third- and fourth-year students mainly study in the classroom, and fifth- and sixth-year students learn by participating in clinical practice. Practice at university hospitals includes lectures, conferences, and participation as a team member. In contrast to the education in Europe in which residents and students are expected to act as educators [5,6], the education in Japan is mainly conducted by attending physicians, and residents and students are usually not expected to act as educators. However, to qualify as attending physicians, a minimum of seven years of clinical experience is required. This results in an age gap between attending physicians and the students/residents [22].

Community-based medical education (CBME) is a practical educational method for primary care and family medicine in which medical trainees learn primary care in medical institutions other than university and tertiary hospitals [23]. In CBME, medical students and residents can acquire practical knowledge, skills, and attitudes that cannot be sufficiently learned at universities and tertiary hospitals, such as clinical reasoning, interprofessional collaboration, and community medicine [24]. Furthermore, owing to the lack of medical resources in rural community hospitals, medical students may alleviate the pressure on healthcare workers in medicine and medical education by participating in patient care as members of medical teams under the framework of legitimate peripheral participation. They can be trained by medical teachers, as well as peer residents and students through the implementation of the RaT framework. Thus, CBME with legitimate peripheral participation, cognitive apprenticeship, and RaT are beneficial to medical students and institutions. For these reasons, the implementation of RaT programs in CBME can have significant effects. Previous reports on RaT were mainly from university and tertiary hospitals. However, it remains unclear whether RaT is beneficial for CBME. Therefore, the research question of this study is: How does the implementation of RaT in CBME in rural community hospitals affect the learning of medical students and residents?

Differences in the educational methods of university and rural community hospitals and low stakeholder motivation for medical education result in learners experiencing difficulties and losing motivation for learning medicine [25,26]. In contrast, RaT can be independent of stakeholder motivation, and medical students can benefit through mutual learning and collaboration. The effectiveness of their learning can be influenced by their learning contexts and environments. By clarifying the effects of RaT in CBME, RaT can be a collaborative education with stakeholders, not only in university and tertiary hospitals, but also in rural community hospitals. Additionally, clarification of the current issues and changes in medical students’ and residents’ perceptions regarding RaT in CBME may lead to the facilitation and quality improvement of its practice in CBME. Therefore, this study aimed to investigate the impact of RaT on students’ learning and their perceived difficulties by interviewing medical students and residents who participated in the RaT programs in CBME at a rural community hospital in Japan.

## 2. Materials and Methods

### 2.1. Participants

The participants of this study were medical students and residents who participated in the CBME curriculum in Unnan City Hospital. All medical students and residents were requested to participate in the study. Participants were informed about the study and told that their participation was completely voluntary and that they could withdraw at any time without affecting their evaluation.

### 2.2. Setting

Unnan City Hospital is a community hospital located in one of Japan’s most remote areas and is used for the implementation of CBME in the prefecture. The city population was 38,882 in 2017, with 37.82% over 65 years of age. The city is roughly a 30-min drive from Izumo City, where the nearest tertiary hospital is located. Fifth- and sixth-year medical students and second-year postgraduate medical residents underwent more than two weeks of training in family and rural medicine at the hospital. There were five physicians specializing in family medicine and primary care, who facilitated the participants’ learning in clinical settings by working with them.

### 2.3. CBME in Unnan City Hospital

During their training, the participants worked with family physicians at the community hospital to learn about the most frequent illnesses and how they were managed through systemic practice and person-centered care, a comprehensive and integrative approach. The participants were given the opportunity to take a history and conduct a physical examination as part of the team and discuss the future treatment direction with the approval of the patient.

They collaborated with paramedics to learn about interprofessional work. To enhance their learning experience, the participants reflected on their performance through 10-to-15-min discussions with their teachers and peers.

### 2.4. Data Collection

One-on-one semi-structured interviews were conducted on the last day of each participant’s CBME training. Each interview lasted approximately 30 min and was conducted by the primary researcher. All interviews were recorded and transcribed verbatim. As an interview guide, five questions were prepared as follows: (1) “Do you have any previous experience with RaT?” (2) “What are the advantages of RaT based on your experience?” (3) “What are the disadvantages of RaT based on your experience?” (4) “How was your relationship with the team members during your training?” (5) “Do you have any suggestions for the improvement of RaT?”. Participant selection, data collection, and data analysis were conducted until theoretical saturation was reached and a complete view of RaT was obtained.

### 2.5. Data Analysis

A thematic analysis using the six steps proposed by Braun and Clarke was conducted [27]. In the first step, two researchers reviewed the transcripts and notes to understand the content. In the second step, the primary researcher generated succinct labels to identify important features of the data. Using open coding, the transcribed data were divided into smaller units, and the codes were labeled based on the interpretations. The primary researcher first labeled the codes and then discussed them with a second researcher to enhance credibility. Then, using axial coding, the relationships between the labeled codes were explored, organized, and grouped to generate concepts. In the third step, the relationships among the concepts were elaborated and developed into themes. In the fourth step, the themes were reviewed and refined by the researchers. In the fifth step, each theme was defined and named. Finally, in the sixth step, the relationships between the themes were explored, and a report was generated. Then, the underlying concepts and themes were investigated. Each step was performed iteratively until a consensus was reached among the research team.

## 3. Result

All three residents and ten medical students who completed the CMBE curriculum agreed to participate and were interviewed over one year. All of the residents were in their second year. Of the ten medical students, three were in their sixth year, and seven were in their fifth year. Approximately 62% (*n* = 8) were male. Not all of the residents were majoring in family medicine.

The analysis revealed three main themes: (1) lack of educational experience with RaT, (2) effectiveness of RaT, and (3) challenges of RaT. Each theme consisted of several concepts. The first theme consisted of three concepts, namely, limited education with a supervising physician, lack of mutual education between near-peers, and prejudice against RaT; the second theme consisted of four concepts, namely, interactive relationships among learners, mutual educational relationship without resistance, learning through feedback, and participating in medical teams; and the third theme consisted of two concepts, namely, transactional distance from senior doctors, and comparison with education systems in university hospitals (Table 1). Since the participants were mainly taught didactically by their supervisors and lacked experience with RaT, they were prejudiced against it. However, after participating in the RaT program in CBME, the participants felt that it greatly enhanced the relationships between learners and educators. Additionally, they felt that they could effectively participate in medical practice as team members through the concept of mutual education in RaT. Furthermore, feedback from their educators and peers positively influenced their attitudes toward knowledge and learning. The differences in the educational methods between RaT and university hospitals (which the participants were familiar with) became barriers to the smooth implementation of RaT in CBME. The participants were concerned that the implementation of RaT in rural community hospitals might negatively influence the learning outcomes of medical students due to lesser participation by senior doctors.

### 3.1. Lack of Educational Experience with RaT

#### 3.1.1. Limited Education with a Supervising Physician

The participants received their medical education, mainly as observers, from supervisors at university hospitals; they were unfamiliar with other types of educational methods. Some of the statements of the participants are as follows: 

“When learning in clinical settings, I mostly consulted the mid-career doctors regarding patients, and not the senior residents.”(Student 1)

“I learned mostly from attending physicians who were older than me and held leadership positions.”(Student 2)

“During my bedside learning, I often learned from attending physicians.”(Student 3)

The participants thought it was evident that the attending physicians were responsible for providing medical education. Furthermore, they recognized that before participating in the CBME, the majority of their medical education was imparted by the attending physicians.

#### 3.1.2. Lack of Mutual Education between Near-Peers

The participants lacked experience in being taught by near-peers in medical training in their previous learning settings. Additionally, they were not given the opportunity to teach their near-peers about clinical issues, and their attitudes toward medical education and the staff were affected by their prejudices. Some of the statements of the participants are as follows: 

“I would ask the staff doctors to check my medical records, and not the junior or senior residents. However, I sometimes hesitated to approach them because they appeared to be occupied.”(Student 3)

“Occasionally, I was taught by doctors who were closer in age to me. But the closest in age were the senior residents, and not the juniors.”(Student 9)

“There were less opportunities to learn from junior residents as there were few junior doctors in my previous learning setting.”(Doctor 2)

The participants stated that they had been taught by senior residents, but not by junior residents. They said that this was because there were fewer junior residents, and they were usually busy. 

#### 3.1.3. Prejudice against RaT

The participants were taught by staff doctors and lacked experience in near-peer learning before participating in the RaT program in the community hospital. Consequently, they considered didactic learning provided by the staff doctors to be more effective than mutual learning among peers and did not recognize RaT as an efficient educational technique. One participant stated:

“I think that I was able to learn more effectively in a one-on-one setting because a more skilled teacher would teach me, unlike in RaT.”(Student 5)

As a result of their previous didactic educational environment, the participants felt that learning from experienced and skilled doctors was more effective than learning from residents. Medical students were prejudiced against RaT, and considered peer learning to be less effective than didactic learning in clinical situations. In the beginning of their training, they were uncertain about the educational environment of RaT.

### 3.2. Effectiveness of RaT

#### 3.2.1. Interactive Relationships among Learners

Since learners and resident teachers shared a similar knowledge base, it was easier for them to ask questions and discuss their experiences. Additionally, the resident teachers were able to understand learners’ difficulties as they shared similar social contexts. They also felt that spending more time with peers and teachers in clinical situations lowered the barriers to communicating with each other. Some of the statements of the participants are as follows: 

“I think I can relate to physicians who are closer in age to me. In contrast, I am reluctant to approach senior doctors because of the significant differences in experiences between us. I realized that it is important to learn from someone who shares similar learning backgrounds.”(Student 1)

“Spending adequate time with my peers and near-peers enabled me to establish good relationships with them, and allowed me to comfortably ask them any questions I had.”(Student 10)

“Mutual learning with near-peers enabled me to understand what junior residents were expected to master in their second year.”(Student 7)

Through RaT, resident teachers and learners enhanced social congruence by spending adequate time together. Additionally, the cognitive congruence, that is, the similar knowledge base and learning contexts that they shared, allowed them to understand each other effectively. Moreover, the students regarded residents as role models by closely experiencing clinical situations with them.

#### 3.2.2. Mutual Educational Relationship without Resistance

Prior to participating in the RaT program, the participants lacked teaching experience. By teaching in CBME, they experienced the joy of teaching others. Additionally, after various teaching situations, they realized that the curricula and teaching methods should be adjusted according to learners’ and teachers’ knowledge and experiences. Some of the participants stated:

“Teaching was difficult. It was challenging to simultaneously accommodate the needs of the students in teaching as they both differed in their characteristics. Not only the two students here, but every student has different needs, so it was challenging to ascertain what they needed.”(Student 1)

“I felt that teaching itself was fun, but I also wondered if I taught something wrong because of my limited knowledge. Therefore, I was motivated to learn more to acquire relevant knowledge and skills.”(Doctor 1)

“One can learn more by teaching than I thought.”(Student 8) 

Moreover, the participants recognized that teaching is learning. They were motivated to increase their knowledge through teaching. Additionally, they wanted to change their perceptions of learning by focusing on the input, as well as learning to teach.

“I think teachers can learn a lot. If we do not understand something, we can research it together.”(Student 4)

The learners believed that solving questions with their teachers was effective for both. Furthermore, the teachers themselves deemed it necessary to improve their teaching skills to enhance their knowledge of teaching as well as clinical medicine.

#### 3.2.3. Learning through Feedback

The participants expressed that they were able to acquire knowledge through discussions of clinical problems and questions with their near-peers, which further contributed to increasing their learning motivation and preparing them to consider new clinical questions. Moreover, they mentioned that the consistent follow-up of their learning by their near-peers increased and sustained their learning motivation and promoted self-learning. Some of the participants’ statements are as follows:

“The teachers cared about me. They encouraged me to think deeply about the clinical problems that I encountered, apart from answering my questions.”(Student 3)

“I had sufficient time to reflect on my daily performance, which enabled me to identify my strengths. Conversely, realizing my weaknesses was emotionally challenging as my future specialty depended on it, which made me critical of myself. However, with the assistance of my near-pears, I was able to stay motivated and study harder.”(Doctor 2)

“I am not a proactive person. However, my teachers consistently encouraged me to be more active through feedbacks, which enabled me to change my attitude toward learning in clinical situations.”(Student 3)

Hence, through feedback and reflection, the participants improved their clinical skills and knowledge. Additionally, they were able to change their attitude from passive participation (passively following instructions) to active participation (actively observing patients and conducting physical examinations) in clinical situations.

#### 3.2.4. Participating in Medical Teams

The participants were satisfied because of their mental safety and the ease of communication with their team members. They could participate effectively in medical teams. Moreover, they communicated with their patients and answered their clinical questions, which contributed to the establishment of trusting relationships between them. 

“All the team members were very approachable, and I could comfortably ask them questions. Moreover, during the study sessions, they taught me many things I did not know. I felt comfortable working with them.”(Doctor 1)

“I spent a lot of time monitoring patients, and one of my roles in the medical team was to observe minor changes in the patients and tell the doctors about it.”(Student 6)

“When I did a procedure, I was entrusted as a team member. For example, when I rotated in surgery, I realized the difference between simply watching the operation and physically participating in it. I had a similar experience when performing Gram stains for the diagnosis of infections.”(Student 9)

The participants who came from outside the hospital felt that mental safety helped them to participate in the medical teams effectively. They were aware of their roles in the teams, as they had adequate experience due to legitimate peripheral participation and cognitive apprenticeship. Additionally, being assigned specific roles gave them a sense of participation in clinical procedures and motivated them to acquire more medical knowledge to contribute to the teams.

### 3.3. Challenges of RaT

#### 3.3.1. Transactional Distance from Senior Doctors

As the participants spent more time with their near-peers, they spent less time with the attending doctors, which mentally distanced them from the doctors. Since they perceived senior doctors to be highly specialized and experienced, they thought that the transactional distance resulted from their lack of participation in RaT. Furthermore, regarding educating near-peers, they compared themselves with the attending doctors and considered themselves incompetent to be teachers.

“I felt distant from the attending physicians when we were in a conference without the resident doctors. This situation could be improved by spending more time with them. The attending physicians did not give us many lectures; however, I think that increasing the number of lectures would help overcome the barriers.”(Student 4)

“In clinical encounters, the attending physicians are more confident because they are more experienced. When dealing with a drug that I have not used before, the experience and confidence of the attending physicians encourages me to use it. Junior doctors do not possess such experience.”(Doctor 1)

“To be honest, there is no perfect teacher. Teaching quality may not always be directly proportional to clinical experience, but I think it is preferable to be taught by experienced medical teachers.”(Student 1)

RaT increased the social and cognitive congruence between the teachers and learners, resulting in effective relationships. However, it increased the transactional distance between the learners and the senior doctors. It became more challenging for them to communicate with senior physicians and decrease the knowledge and skills gap between them. The participants also felt that their transactional distance from attending doctors decreased their training quality.

#### 3.3.2. Comparison with Education Systems in University Hospitals

The participants felt that the disease prevalence, educational facilities, and education systems differed between rural and university hospitals. At university and general hospitals, students were in charge of one patient, while in community hospitals, students examined many patients. In terms of facilities, university hospitals had simulation equipment and education management systems to store study materials for learners, which they believed contributed to efficient learning. Additionally, while students were assigned to one supervising physician at university hospitals, they were taught by a team at community hospitals, which they thought could create problems in educational responsibilities. 

“We (university students) use Webex and others; however, they (professors) provide slides which are converted into PDF files and uploaded on the Internet for the students to download. Moreover, they exclusively create videos for students to watch. I think it would be great to have such a service.”(Student 2)

“I feel that I did not get enough time to observe the patients.”(Student 6)

“I was worried that I would have to meet the doctors repeatedly and whether or not it was acceptable… At times, I did not know which doctor checked my patient.”(Doctor 2)

The participants expressed that the learning experience of general diseases and practices in a clinical setting could not be experienced in a university setting. The experiences that they gained changed their perceptions of clinical reasoning and general practice. They also pointed out that the differences in the facility systems could contribute to inefficient learning, and that the team-teaching systems could make the educational responsibilities for the doctors vague. Moreover, it could result in students feeling neglected in some situations. 

## 4. Discussion

This study revealed the effects and challenges of the implementation of RaT in rural community hospitals on medical students’ and residents’ learning. The findings suggest that the participants were prejudiced against RaT because they received limited education from their attending physicians at university hospitals and lacked educational experience through RaT. However, after participating in RaT, the participants felt that it facilitated the establishment of good relationships between learners and teachers. Additionally, owing to their effective relationships, they were able to comfortably participate in medical teams. Moreover, through feedback and discussions, the participants observed changes in their attitudes toward medical knowledge and learning methods, and recognized their roles in medical teams. Furthermore, the learners expressed that the differences in the education systems in university hospitals could be a barrier to the smooth implementation of RaT, and that the lack of participation of senior doctors might lessen the impact of RaT on students’ learning.

The participants recognized the challenges of teaching and learning in RaT, but they also felt that their teaching skills and learning were considerably improved through mutual education and constant reflection among near-peers. Internationally, the teaching and supervision of peers and students are recognized as essential competencies for residents. In addition to the growing recognition that residents have an essential role in education, medical educational facilities have started to actively train residents to become teachers and promote the RaT program implementation [28]. Conversely, in Japan, attending doctors are responsible for teaching the residents. To qualify as attending physicians, a minimum of seven years of clinical experience is required. Consequently, it results in a transactional distance between attending physicians and the students/residents. However, in the guidelines for residency training issued by the Ministry of Health, Labor, and Welfare (MHLW) [22], “senior physicians” are defined as physicians with more clinical experience than residents. It further states that senior physicians play a crucial role in educating residents/students. An increase in the participation of senior physicians in medical education might increase educational opportunities for the implementation of RaT.

RaT in CBME can be implemented effectively in Japanese contexts, which can facilitate the reasonable allocation of physicians. In the present study, the effect of RaT was evaluated in a Japanese community hospital setting. Since there are few medical educators and specialists in remote areas in Japan, medical education is mainly provided by university and tertiary hospitals [29]. In medical education at university hospitals, learners have fewer opportunities to experience primary care and community medicine [17]. Similar to the effects of RaT in Europe and the United States, as verified in previous studies [2,12], this study revealed that the Japanese residents and students believed that RaT created “interactive relationships among learners” and “mutual educational relationships,” and that the relationships facilitated “learning through feedback.” Furthermore, the learners could identify their roles in and contribute to the medical teams. In Japan, the localization of physicians is a significant challenge. Previous studies have shown that time constraints in medicine and increasing faculty obligations drive the need for RaT [7] to maintain teaching quality, even in situations with limited medical resources [8]. Training in CBME can motivate learners to work in communities and remote areas [29,30]. By increasing the quality of CBME, RaT has the potential to motivate more physicians to work in communities, providing a solution to the problem of the localization of physicians.

In RaT in CBME, the effective involvement of attending physicians can improve the learning quality of students. In this study, the participants thought that the quality of learning decreased due to the decrease in the number of attending doctors teaching students. Despite some studies reporting the direct impact of resident teaching on learner performance, objective evidence is lacking [1]. The participants’ opinions might be due to the didactic educational culture of Asian countries, where people tend to think that those with more knowledge and experience are better teachers [31]. This cultural aspect has been reported in settings of problem-based learning (PBL) [31], but has not been reported in RaT. This study revealed that residents and students viewed RaT favorably and changed their learning attitudes. Future research is needed to understand changes in knowledge and skills. The participation of attending physicians would lead to an improvement in the educational quality of RaT in CBME. In previous reports, teachers complained about overburdened educators [13], but this was not mentioned by the participants in this study. This might be because the responsibilities were distributed among all of the team members. Further interviews with attending physicians are needed.

RaT may have a high affinity for CBME in remote areas where medical resources are lacking. Compared to learners in European countries, in Asian countries, including Japan, learners are reluctant to express their views, and avoid critical discussions [32]. These Asian cultural attributes have been reported as potential barriers for some educational methods, such as PBL [31]. However, in this study, the learners mitigated these barriers because they shared social and cognitive congruence with the teachers. They tried to solve clinical problems through collaboration with their peers. By managing learning contexts and environments, RaT in CBME can be an effective teaching method for learners in Asian countries. Furthermore, since many Asian cultures tend to take a more holistic rather than an individualistic approach to life [32], RaT will be more effective in Asia when it is implemented in medical teams, as was done in this study. As the present CBME lacks evidence regarding the effects on students’ qualitative and quantitative learning, further studies should be conducted to investigate the effectiveness of rural CBME in different contexts [33].

This study has several limitations. First, this study’s participants were limited to medical students and residents in remote areas of Japan. However, since education is standardized according to the model core curriculum established by the MHLW, transferability was maintained. Second, there could be a problem with credibility; as the students chose their training site, we could only investigate learners about general medicine in a rural community hospital. In North America and Europe, it is well established that residents do actively teach medical students and junior trainees, and this is an expectation [34,35]. We hope that this concept will be expanded to the whole of Japan and further spread in the Asian context. In the future, conducting similar studies on medical student residents regarding RaT in Asian contexts may reveal different issues in medical education for the effective application of RaT with respect to educational culture. Third, the interviewer held an educational position at the community hospital and taught the participants. This may have introduced bias into the participants’ responses to a certain extent and may reduce confirmability, and the participants might have felt coerced to participate in the study. However, as the interviewer was in close contact with students and residents through the RaT, the participants may have expressed their views more comfortably and honestly. 

## 5. Conclusions

This study revealed the effects and challenges of the implementation of RaT in rural community hospitals on medical students’ and residents’ learning. The participants were prejudiced against RaT; however, after experiencing it, they felt that it helped establish good relationships between learners and teachers. Additionally, owing to their effective relationships, they were also able to participate in medical teams comfortably. The greater participation of senior doctors might strengthen the learning effect of RaT. RaT in rural CBME should be applied more frequently in various rural contexts, and its effectiveness should be investigated both qualitatively and quantitatively.

## Figures and Tables

**Table 1 ijerph-18-12410-t001:** The results of the thematic analysis.

Theme	Concept
Lack of educational experience with RaT	Limited education with a supervising physician
	Lack of mutual education between near-peers
	Prejudice against RaT
Effectiveness of RaT	Interactive relationships among learners
	Mutual educational relationship without resistance
	Learning through feedback
	Participation in medical teams smoothly
Challenges of RaT	Transactional distance from senior doctors
	Comparison with education systems in university hospitals

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
