# Peer review of "Effect of Residents-as-Teachers in Rural Community-Based Medical Education on the Learning of Medical Students and Residents: A Thematic Analysis"

_ijerph, 2021, doi:10.3390/ijerph182312410_

Round 1
Reviewer 1 Report
This is an interesting and relevant topic in relation to workforce recruitment and rural medical education. My main concerns are with the introduction and inaccurate portrayal of RaT as a theoretical framework. It is a method of teaching very much based on the social constructivist work of Vygotsky and the work you have cited here in relation to near-peers.
The authors have made several mentioned throughout the transcript to the idea of mental distance. These ideas are better phrased as "transactional distance" (See Moore, 1997, Moore, M. (1997). Theory of Transactional Distance. In Keegan, D. (1997). (Ed.). Theoretical Principles of Distance Education. Routledge,).
I am also concerned by the mis-representation of Grounded Theory in the methods. What has been described is really a thematic analysis, not grounded theory. There is also no reference back to how the researchers ensured credibility, validity and reliability of their findings, trustworthiness etc through member-checking. There was further no discussion about how many instances there was disagreement about the codes. This reporting of the qualitative data analysis needs to be tightened up.
I disagree with the point made, line 84, that "RaT can be a stakeholder-independent way of education" - there is no such thing and this does not make any sense. Education involves learning and participation, which by its very nature involves a stakeholder - whether that be the instructor, the student, the patient, the hospital setting, the community, etc.
In the Materials and Methods it was not clear to me how participants were recruited an how coercion was avoided. I note that informed consent was provided but it was not clear if any of the researchers were involved in teaching or assessment and so students may have felt coerced.
In section 2.1 it would have been nice to know how many medical students participated in the CBME curriculum, is 10 a representative sample?
In section 2.2 it would be nice to know more about Unnan - what is the population? how remote is it? where is the nearest tertiary site? How many students are based there at any one time? What does the RaT involve specifically? Is there any structure?
Were the participants male or female? in 5th or 6th year?
The main findings and conclusions were well written and some good points are made.
Reviewer 2 Report
Thank you for asking me to review this study of residents as teachers in a Japanese rural primary care educational program. Dr. Nishikura and colleagues conducted a qualitative study of medical students and residents training at Unnan City Hospital, Japan using semi-structured interviews and a grounded theory approach to study design and data analysis. This study provides important information about rural training in Asia, which is quite under-represented in the medical education literature. The paper is generally well written and helps fill this gap in the literature. I do have some suggestions for revision:
Major comment – Introduction: Can the authors provide some background in the Introduction about the current state of residents as teachers in Japan? Are residents generally expected to teach medical students/junior residents/other health care trainees? If so, does this teaching generally occur in the context of clinical work (informal education) or classroom learning (formal education through lectures, conferences, etc)?
Major comment – Introduction: It would be helpful in the Introduction if the authors could provide some context about how students and residents learn in Japan. In North America and Europe, first and second year medical students learn primarily in the classroom, whereas senior medical students and residents learn primarily through participation in clinical practice1,2 For senior medical students and residents, classroom learning such as academic half days or noon conference remains and important but complementary aspect of their education and identity formation.3,4 Some general description of the curriculum for students and residents in Unnan would be helpful for the reader to frame the current study.
Minor comment: In the Discussion, paragraph 2, the authors mention that attending physicians must have seven years experience to qualify as teachers. This seems to be an important piece of information to put this study in context and I would suggest moving it to the Introduction, as one of the reasons for doing the present study.
Minor comment – Methods: Please clarify if the residents in the study were all in a Family Medicine program, and what year of residency they were in.
Major comment – Methods: It is not clear whether the study involved training residents to be teachers, or simply solicited the opinions of medical students and residents about residents teaching.
Major comment – Discussion: In North America and Europe, it is well established that residents do actively teach medical students and junior trainees, and this is an expectation.5,6 It would be helpful if the authors describe how they envision their next steps to develop RaT in their context, and how their findings may be transferable to other centers that are also seeking to develop RaT programs.7
- Teunissen PW, Scheele F, Scherpbier AJJA, et al. How residents learn: qualitative evidence for the pivotal role of clinical activities. Med Educ 2007; 41(8): 763-70.
- Billett S. Learning through health care work: premises, contributions and practices. Med Educ 2016; 50(1): 124-31.
- Chen LY, McDonald JA, Pratt DD, Wisener KM, Jarvis-Selinger S. Residents' views of the role of classroom-based learning in graduate medical education through the lens of academic half days. Acad Med 2015; 90(4): 532-8.
- Chen LYC, Hubinette MM. Exploring the role of classroom-based learning in professional identity formation of family practice residents using the experiences, trajectories, and reifications framework. Med Teach 2017; 39(8): 876-82.
- Busari JO, Scherpbier AJJA, Van Der Vleuten CPM, Essed GGM. The perceptions of attending doctors of the role of residents as teachers of undergraduate clinical students. Medical Education 2003; 37(3): 241-7.
- Ratan BM, Johnson GJ, Williams AC, Greely JT, Kilpatrick CC. Enhancing the Teaching Environment: 3-Year Follow-Up of a Resident-Led Residents-as-Teachers Program. Journal of Graduate Medical Education 2021; 13(4): 569-75.
- Bree KK, Whicker SA, Fromme HB, Paik S, Greenberg L. Residents-as-Teachers Publications: What Can Programs Learn From the Literature When Starting a New or Refining an Established Curriculum? Journal of Graduate Medical Education 2014; 6(2): 237-48.
